# Evaluation of Inbreeding in the Slovak Simmental Breed and Its Effect on Length of Productive Life

**DOI:** 10.3390/ani14121811

**Published:** 2024-06-18

**Authors:** Eva Strapáková, Peter Strapák

**Affiliations:** 1Institute of Nutrition and Genomics, Faculty of Agrobiology and Food Resources, Slovak University of Agriculture in Nitra, Trieda Andreja Hlinku 2, 949 76 Nitra, Slovakia; 2Institute of Animal Husbandry, Faculty of Agrobiology and Food Resources, Slovak University of Agriculture in Nitra, Trieda Andreja Hlinku 2, 949 76 Nitra, Slovakia; peter.strapak@uniag.sk

**Keywords:** inbreeding, Simmental cows, length of productive life, survival analysis, Weibull model

## Abstract

**Simple Summary:**

Inbreeding is the mating of related individuals. Inbreeding has been used in the past to create new breeds. Individuals selected for inbreeding were those who had acquired the genes of their parents, leading to higher performance in a particular trait. However, the genetic and phenotypic effects of inbreeding can dramatically affect the herd. An inbred individual is more likely to be homozygous for any gene, so the animal may also show undesirable traits and a reduction in average phenotypic performance, which is called inbreeding depression. Inbreeding depression negatively affects mainly reproductive traits, fertility, growth parameters, and longevity. Inbreeding depression is essentially the opposite effect of heterosis, which is the result of crossing lines or breeds.

**Abstract:**

This study aimed to estimate the average inbreeding coefficient in Slovak Simmental dairy cattle and evaluate the effect of inbreeding on the length of productive life. All pedigrees included 463,282 animals dating back to 1914. The inbreeding coefficients for each animal in the pedigree were computed using the software CFC 1.0. Length of productive life (LPL) was defined as the time (days) from the first calving to culling, death, or censoring. The influence of inbreeding on the length of productive life was calculated and tested using the Weibull proportional hazards model. The average inbreeding coefficient, the average number of discrete generation equivalents, and the average longest ancestral path for inbred animals were 0.01, 6.59, and 13.08, respectively. While the largest decrease in the mean coefficient of inbreeding was observed from the year of birth 1995 (F = 1.50%) to 2001 (F = 0.59%), an increasing trend of inbreeding in the population was found from 2003 onwards. A weak but significant effect of inbreeding on the length of productive life of Simmental cows was confirmed using survival analysis.

## 1. Introduction

Inbreeding is generally a reserved term for mating animals that are more closely related than the breed average. Inbreeding leads to a loss of genetic variability and an increase in the frequency of undesirable deleterious alleles, which can cause reproductive and productive problems in animals [1,2]. Most animals carry undesirable genes that usually remain hidden unless the animal is homozygous. Inbreeding does not create undesirable recessive genes but tends to accentuate these unfavorable genetic traits. This leads to a decline in average phenotypic performance called inbreeding depression. Inbreeding depression reduces the expression of dominant homozygous alleles [3] and reduces the rate of response to selection for selected traits [4]. Genetic diversity enables genetic improvement of animal production traits and, therefore, is important both economically and environmentally [5]. On the contrary, increasing the inbreeding coefficient by 1% resulted in a reduction in milk, fat, and protein production by 11.99 kg, 0.39 kg, and 0.29 kg, respectively [6]. García-Ruiz et al. [7] stated that at each percentage point of increase in inbreeding, milk, fat, and protein production decreased by 88 kg, 3.16 kg, and 2.57 kg, respectively. Each 1% increase in inbreeding reduces lifetime net income by $22 to $24 [8]. Thompson et al. and Gorelik et al. [9,10] found a significant loss of productivity and survival that was caused by increased inbreeding rates. The greatest production losses occurred early in life and in the early stages of lactation. The shortened production life probably had a greater negative impact on dairy farm economics than production losses. The authors [9,10] confirmed that production lifespan was higher in outbred and inbred cows with moderate levels of inbreeding compared to cows with higher inbreeding coefficients. A slight trend toward a higher relative risk of culling in more inbred animals was also found by Sewalem et al. [11]. The negative effects of inbreeding that represent some economic loss must be weighed against the additional genetic gain that would be expected from each mating [12]. More recently, pedigree information has been rigorously analyzed to avoid mating of related individuals. Nevertheless, inbreeding still occurs because a relatively small number of elite bulls are sires of these animals [13]. Traditionally, pedigree-based inbreeding coefficients have been used to manage the inbreeding that occurs within a population [14]. Assessing the impact of factors on longevity is done by survival analysis using the Weibull model, which correctly accounts for both censored data and skewed distributions of survival data [15].

The objective of this study was to evaluate the pedigree structure, calculate the average inbreeding coefficient, and estimate the effect of inbreeding on longevity in Slovak Simmental dairy cattle.

## 2. Materials and Methods

Data for this study were obtained from the Slovak Breeding Services, s.e. (Bratislava, Slovakia). The level of inbreeding and its effect on longevity was studied in a pedigree file of Slovak Simmental cows. The data consisted of 244,252 cows born between 1995 and 2021 from 1045 herds and 2583 sires.

All pedigrees included 463,282 animals dating back to 1941, of which 12,305 were males and 450,977 were females. For the pedigree analysis, the software package CFC 1.0 (Contribution, Inbreeding (F), Coancestry) [16] was used. CFC contains a set of programs to compute inbreeding coefficients, assess relationships between individuals, optimize matings to minimize the average inbreeding in the next generation, and compute probabilities of gene origin. CFC also offers useful information on the structure of the pedigree, checking the pedigree for possible errors, etc. [16]. The inbreeding coefficients for each animal in the pedigree were computed using a modified version of Colleau’s algorithm [17,18]. The average inbreeding coefficient was calculated for the inbred individuals in the pedigree. Subsequently, the number of inbred cows in the evaluation dataset was determined, and the average inbreeding coefficient by year of birth was calculated. The percentage of animals with complete pedigree through the fifth generation is shown in Figure 1.

Length of productive life (LPL) was defined as the time (days) from the first calving to culling, death, or censoring. Cows that met the following conditions were included in the evaluation of length of productive life:-date of birth and first calving indicated-age at first calving between 600 and 1200 days-known sire of the cow.

The lifetime records were considered uncensored (complete) for cows that died or were culled at the time of analysis. Data from cows that were alive at the end of the analysis, that reached more than five lactations, that had a milk yield of less than 1700 kg, that were removed from the Slovak National Milk Recording System, that were sold for dairy purposes, and that were from herds with less than ten calvings per year were considered censored. The database was prepared using SAS 9.2, Enterprise Guide 4.2. [19]. Survival analysis [20] was used to assess the impact of factors on the length of productive life, which offered the possibility to include incomplete information (censored data) [20]. In addition, longevity data has no normal distribution. The following Weibull proportional hazard model was used to evaluate the effect of selected factors on the length of productive life of cows.
λ(t) = λ0 (t) exp(P + M + herd + age + F + HYS + sire)
where: t is time in days from the first calving to culling or censoring data; λ(t) is the hazard function of a cow at time t; λ0(t) is the Weibull baseline hazard function with scale parameter λ and shape parameter ρ; P is the fixed time-dependent effect of parity (5 classes with changes at each calving year from 1996 to 2023); M is the fixed time-dependent effect of the milk yield expressed as within herd-year standard deviations (SDs; five classes: M > +2 SD from the herd-year average, 1 SD ≤ M ≤ +2 SD, −1 SD < M < +1 SD, −2 SD ≤ M ≤ −1 SD, M < −2 SD); herd is the fixed time-dependent effect of the annual change in herd size (five classes: −5% ≤ herd ≤ +5%, −30% < herd < −5%, herd ≤ −30%, +5% < herd < +30%, herd ≥ +30%); age is a fixed time-independent effect of the age at first calving (5 classes: 600–720 days, 721–840 days, 841–960 days, 961–1080 days, 1081–1200 days); F is a fixed time-independent effect of the inbreeding coefficient (6 classes: F = 0, 0 < F ≤ 0.01, 0.01 < F ≤ 0.03125, 0.03125 < F ≤ 0.0625, 0.0625 < F ≤ 0.125, F > 0.125); HYS is the random time-dependent effect of the herd × year of calving × season of calving interaction, following a normal distribution with change points on April 1 and October 1 in each calendar year (variance = 0.38); and sire is the time-independent random effect of the sire of the cow, assuming to follow a normal distribution with variance of 0.02. The paternal part of the pedigree included 5111 bulls. The effects of the factors were tested by the Sequential Likelihood Ratio Test.

## 3. Results

Table 1 shows the basic pedigree structure of 244,252 Slovak Simmental cows with 463,282 individuals in pedigrees. The average inbreeding coefficient (F) in the inbred individuals was slightly higher for males (F = 0.018) compared to females (F = 0.011). The total number of founders was 66,813.

Descriptive statistics for the F, the number of discrete generation equivalent (GE), and the longest ancestral path (LAP) for inbred animals in the pedigree are presented in Table 2. The average inbreeding coefficient was 0.01 for inbred animals, and the maximal F was 0.39. Ancestors were traced back to 1941; the longest ancestral path reached a maximum of 21.

The highest percentage (99.26%) of inbred animals reached an F up to 0.10. The inbreeding coefficient in the range of 0.31–0.40 was reached by only nine animals (Table 3).

The mean inbreeding coefficient was calculated separately for inbred cows (n = 161,919) in the base set. The trend of the average inbreeding coefficients over time is illustrated in Figure 2. The largest decrease in the average inbreeding coefficient was observed from the birth year 1995 (F = 1.50%) to 2001 (F = 0.59%). Since 2003, an increasing linear trend of inbreeding in the population has been detected. It reached the highest average value in cows born in 2021 (F = 1.73%).

Figure 3 shows the percentage of cows by classes of inbreeding. Of the total, 33.71% were non-inbred cows, and 44.05% of the cows achieved an inbreeding coefficient of up to 1%. The group of animals with inbreeding coefficients ranging from 6.26 to 12.5% made up a 0.93% proportion of the population.

The highest proportion of non-inbred cows (82.90%) by year of birth was found in 1995 to 1999 (Table 4). In the group of inbred cows with up to 1% inbreeding, the highest proportion was calculated for the years 2010 to 2014 (63.83%). In groups 1 < F ≤ 3.125 and 3.125 < F ≤ 6.25, a linear increase in the percentage of inbred cows was found from 1.81% to 48.89% and from 0.34% to 4.26%, respectively. In group F > 12.5%, there was an increase of 0.14% in cows over the entire observation period (Table 4).

The inbreeding coefficient increased by 0.7% between 2000 and 2004 and 2015 and 2021 (Table 5). The average number of discrete generation equivalents and the longest ancestral path also showed an increasing trend from 4.89 to 7.9 and from 10.15 to 15.61, respectively. The average change in the inbreeding coefficient between 1995 and 1999 (−0.16%) confirms the downward trend of F over this period. On the contrary, in the last period under study, the trend of F was upward (0.10%) (Table 5).

The influence of factors was tested using the sequential likelihood ratio test (Table 6). Milk production had the highest effect (204,784.9 ***) on the length of productive life. The test confirmed the weakest (91.606***) but significant effect of inbreeding on the length of productive life.

Figure 4 displays the relative risk ratio of Slovak Simmental cows according to the degree of inbreeding. Cows with an inbreeding coefficient of up to 1% were at 1.032 times higher risk of culling than non-inbred cows. Results in the 6.25 < F ≤ 12.5 and F > 12.5 groups were affected by very few observations. The trend shows a reduction in the productive life for inbred cows compared to non-inbred cows.

## 4. Discussion

Inbreeding is one of the factors observed in crossbreeding and animal breeding. When evaluating inbreeding, it is important to have correct and as complete as possible information about the ancestry of an individual, preferably within several generations. It is reasonable to expect that more complete pedigree information will produce more accurate estimates of inbreeding [21]. Complete pedigree up to the first generation in the evaluated set was achieved by 99.11% of the Slovak Spotted cows, up to the second generation by 87.92% (Figure 1). The pedigree analysis revealed an average number of discrete generation equivalent of 2.94 (Table 1). This parameter constitutes an indicator of the pedigree depth for animals with known parents [22]. A higher average number of discrete generation equivalents was reported by Silva et al. [23] (3.38), Gonzáles-Recio et al. [24] (5.68), Wirth et al. [25] (5.78), and Croquet et al. [1] (6.00).

An average inbreeding coefficient of 1.10% was calculated in the Slovak Simmental inbreed animals (Table 1), which is relatively low compared to the results of other authors. [11] reported that Jersey, Ayrshire, and Holstein breeds achieved inbreeding coefficients of all individuals of 3.6, 3.99, and 3.20%, respectively. Inbreeding in Holstein breed inbred cows in Brazil reached 6.02% [23], inbreeding in the American Jersey breed was 4.6% (F = 4.6%) [9], and inbreeding in the Finnish Ayrshire population was 2% [26]. Average inbreeding coefficients for the Danish Holstein, Jersey, and Danish Red breeds have been reported at 3.9%, 3.4%, and 1.4%, respectively [27], and an inbreeding coefficient of 1.68% has been reported for the Iranian Holstein [6]. High average inbreeding coefficients (9.24% and 5.35%) were found in inbred animals of the Lithuanian Red_pure and Black and White_pure populations [28]. The average inbreeding coefficient of the German Holstein population between 1960 and 2008 was 3.3% [29], and that of the Mexican Holstein population reached 2.6% [7]. [30] reported an inbreeding coefficient of 4.85% for the South African Jersey breed. The mean individual coefficient of inbreeding in German Brown cows was 1.8% [25]. The lower inbreeding coefficient in the population of Slovak Simmental cattle is probably due to the large number of proven bulls used in breeding, where it is possible to mate unrelated individuals using mating programs.

In recent years, an increasing trend of inbreeding in cattle populations has been found. From 2003 to 2021, inbreeding increased by 1.11% in the evaluated population of Slovak Simmental cows (Figure 2). The increase in inbreeding from 1996 to 2013 (+3.7%) was also reported by Hofmannová et al. [2]. The level of inbreeding in the Latvian Brown and Latvian Blue populations tended to increase by the year 2019, reaching 2.61% and 5.20%, respectively [31]. In the German Brown cow population, the increase in the inbreeding coefficient from 1990 to 2001 was 1.3% to 2.4% [25], and for the Holstein, Montbéliarde, and Normande breeds in France, inbreeding increased by up to 4.5 to 5% [32]. The upward trend in the inbreeding coefficient may also be caused by the greater number of younger animals with known pedigrees up to a few generations. The oldest animals have fewer known ancestors; therefore, the inbreeding coefficient is affected by a lack of information, and its calculation may be biased [33]. Such animals may have F = 0, but in reality, this may not be true.

In this study, the average annual rate in 1995–1999 decreased (ΔF = −0.16%). The Slovak Simmental breed had controlled the rate of increase in the inbreeding level best in 2000–2004 at −0.01% (Table 4). This inbreeding trend in this period was caused by the purchase of insemination batches of a large number of proven bulls, mainly from Germany and Austria, which were not related to the population of the Slovak Simmental breed. The selection of unrelated bulls for breeding may also have influenced the increase in genetic variability of the progeny. The Ayrshire and Jersey breeds in Canada achieved a rate of inbreeding of −0.04% and +0.07% per year, respectively, from 2000 to 2010 [34]. Similar results were published by Maiwashe et al. [35], who calculated ΔF per year for the Jersey breed (0.07%) and the Ayrshire and Guernsey breeds, both of which had the same rate (0.05%). In Canadian dairy cattle, the rate of inbreeding was observed for Jersey, Ayrshire, and Holstein breeds from 1980 to 2004 at 0.08%, 0.11%, and 0.17% per year [11]. The higher rate of inbreeding from 2000 to 2007 (0.31% per year) compared to our study was reported by Rokouei et al. [36] in the Iranian Holstein breed.

Inbreeding influences the increase in the proportion of the homozygous population [3]. A recessive form of homozygosity can cause so-called inbreeding depression, which negatively affects animal production, reproduction, and other functional traits. Across all studies and traits, inbreeding had an unfavorable effect on all groups of traits (reproduction/survival, weight/growth, conformation, production, health, and other traits) [37], a 1% increase in inbreeding was associated with a decrease in the median phenotypic value of 0.13% of the trait mean or 0.59% of the trait standard deviation.

In this study, a significant effect of inbreeding on the length of productive life was found (Table 6). Cows with an inbreeding coefficient of up to 6.25% were at a higher risk of culling than non-inbred cows (Figure 4). These results were confirmed by the study of Thompson et al. [9], who found that the average probabilities of survival decreased with increasing inbreeding levels for all lactations studied. Decreased longevity estimates within the herd and year indicated lower production of inbred cows compared to non-inbred cows. The most visible negative impact of inbreeding (F > 10%) on survival in US Jersey cattle at 48 months of age or later was reported also by Nascimento et al. [38]. [30] found that a 1% increase in inbreeding decreased longevity at first lactation by an average of 0.14% and at second lactation by 1.70%. A slight trend toward a higher risk of culling among more inbred animals was stated also by Caraviello et al. [39]. According to the authors, the relative risks of culling for cows with inbreeding coefficients greater than 8% were from 1.05 to 1.1 times higher than the culling risk for cows with inbreeding coefficients of 7% or lower.

## 5. Conclusions

The coefficient of inbreeding of Slovak Simmental cattle has been gradually increasing since 2003. Similar trends have been observed recently in almost all dairy and dual-purpose cattle populations worldwide. Inbreeding negatively affects reproductive and production traits, leading to a shortened production lifespan and ultimately negatively affecting farm economics. The results of the study confirmed the significant effect of inbreeding on the length of productive life in the Slovak Simmental cattle. A reduction in the rate of inbreeding in the population can be achieved by consistent identification of related individuals, e.g., by using computerized mating programs for dairy cows, their complete pedigree information up to at least the fifth generation, the using of genomic proven bulls and by avoidance of remating of related individuals. Inbreeding should only be considered if the increased performance in a given trait significantly exceeds the losses due to inbreeding depression.

## Figures and Tables

**Figure 1 animals-14-01811-f001:**
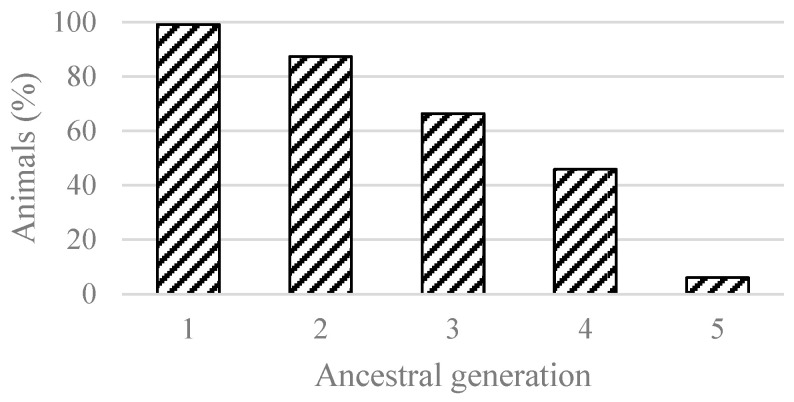
Pedigree completeness up to the fifth generation of ancestors.

**Figure 2 animals-14-01811-f002:**
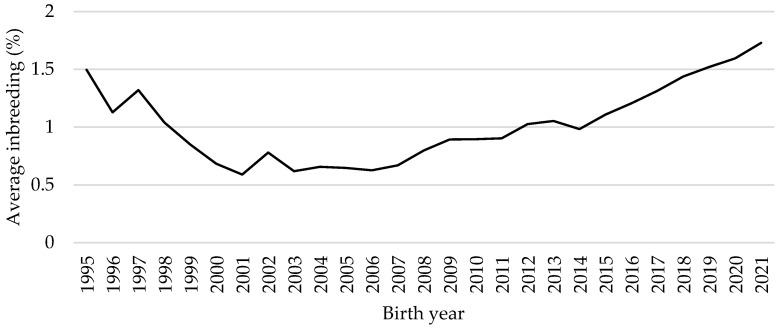
The inbreeding trend in inbred cows by birth year.

**Figure 3 animals-14-01811-f003:**
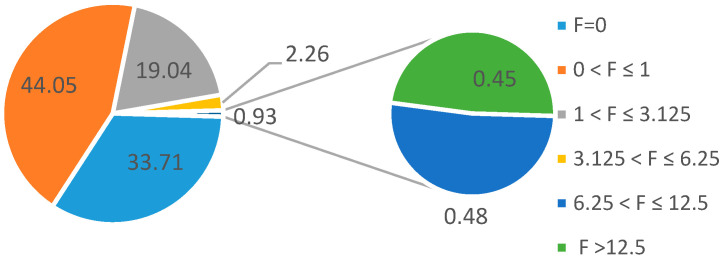
Percentage of cows by classes of inbreeding (%).

**Figure 4 animals-14-01811-f004:**
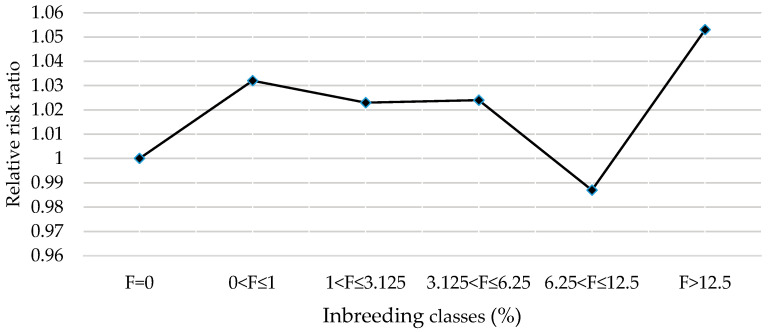
Relative culling risk for inbreeding classes of Slovak Simmental cows.

**Table 1 animals-14-01811-t001:** Pedigree structure.

	Male	Female	Total
Number of individuals	12,305	450,977	463,282
Number of inbred animals	3562	215,695	219,257
Number of founders	2548	64,229	66,813
Average inbreeding coefficient	0.0051	0.0053	0.0051
Average inbreeding coeff. in the inbred animals	0.018	0.011	0.011
Average number of discrete generation equiv.	4.57	4.62	2.94

**Table 2 animals-14-01811-t002:** Average F, GE, and LAP for inbred individuals in pedigree.

Variable	Mean	Std Dev	Minimum	Maximum	N
F	0.01	0.02	2.38419 × 10^−7^	0.39	219,257
GE	6.59	1.15	1.75	9.61	219,257
LAP	13.08	2.21	2.00	21.00	219,257

Mean—average; Std Dev—standard deviation; N—number of inbred individuals.

**Table 3 animals-14-01811-t003:** Distribution of inbreeding coefficients.

Classes of Inbreeding Coefficients	Number of Animals
0.00 < F ≤ 0.05	215,161
0.05 < F ≤ 0.10	2485
0.10 < F ≤ 0.15	671
0.15 < F ≤ 0.20	30
0.20 < F ≤ 0.25	354
0.25 < F ≤ 0.30	547
0.30 < F ≤ 0.40	9

**Table 4 animals-14-01811-t004:** Percentage of cows in each class of inbreeding by birth year.

Birth Year	Class of Inbreeding (%)					
	F = 0	0 < F ≤ 1	1 < F ≤ 3.125	3.125 < F ≤ 6.25	6.25 < F ≤ 12.5	F > 12.5
1995–1999	82.90	14.37	1.81	0.34	0.20	0.38
2000–2004	25.09	37.51	3.97	0.93	0.21	0.28
2005–2009	23.23	62.78	10.78	2.16	0.48	0.56
2010–2014	8.45	63.83	23.15	3.22	0.86	0.49
2015–2021	1.82	43.91	48.89	4.26	0.60	0.52

**Table 5 animals-14-01811-t005:** Average inbreeding coefficient (F), number of discrete generation equivalent (GE), longest ancestral path by birth year (LAP), and annual changes in inbreeding coefficient (Change) of inbred Slovak Simmental cows (N = 161,919) for each period.

Birth Year	N	F		GE		LAP		Change (%)	
		Mean	Std Dev	Mean	Std Dev	Mean	Std Dev	Mean	SE
1995–1999	8976	0.011	0.037	4.89	0.54	10.15	1.19	−0.16	0.12
2000–2004	17,037	0.006	0.020	5.48	0.56	11.15	1.19	−0.01	0.08
2005–2009	36,756	0.007	0.019	6.17	0.59	12.36	1.09	0.06	0.03
2010–2014	43,029	0.009	0.018	6.87	0.58	13.66	1.06	0.02	0.04
2015–2021	56,121	0.013	0.017	7.90	0.65	15.61	1.16	0.10	0.01

N—number of inbred cows; Std Dev—standard deviation; Mean—average; SE—standard error.

**Table 6 animals-14-01811-t006:** Sequential likelihood ratio test.

Variable	Total DF	CHI^2^	R^2^ of Maddala
No covariate	2		
Milk yield classes	7	204,784.9 ***	0.5676
Parity	11	9798.3 ***	0.5846
Herd size classes	16	20,827.0 ***	0.6186
Age at first calving classes	20	800.22 ***	0.6198
Sire	4716	6404.1	(RANDOM)
Inbreeding coefficient classes	4721	91.606 ***	0.6298
Herd-year-season	27,945	45,647.0	(RANDOM)

*** *p* < 0.05.

## Data Availability

These data are subject to disclosure restrictions. Data were provided by the Slovak Breeding Services, s.e.

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
