# Peer review of "Evaluation of Inbreeding in the Slovak Simmental Breed and Its Effect on Length of Productive Life"

_animals, 2024, doi:10.3390/ani14121811_

Round 1

Reviewer 1 Report

Comments and Suggestions for Authors

Manuscript ID:animals-3055636

Title:Evaluation of inbreeding in teh Slovak Simmental breed and its TEMP effect on length of productive life.

General Comments: Inbreeding depression is common in cattle or other livestocks. It negatively affects mainly reproductive traits, fertility, growth parameters, and longevity. This study aim to evaluate the pedigree structure, calculate the average inbreeding coefficient, and estimate the effect of inbreeding on longevity in Slovak Simmental dairy cattle. As a result, the authors found the coefficient of inbreeding of Slovak Simmental cattle has gradually increased since 2003.  Inbreeding affects reproductive and production traits, leading to a shortened production life and ultimately negatively affecting farm economics. These results could provide useful information for cattle production.

Major concerns:

1.Abstract section is missing.

2.In Discussion section OR in conclusion section. It is better to indicate that how cattle production can benifit from these results.

3.Some pedigree data can date back to 1941. Sure, this is a very good material. But, how to make sure these pedigree data are correctly recorded? 

4.The content of this manuscript is not enough for a article. Instead, Communications may be more suitable for this manuscript.

Minor concerns:

The resolution of Figure2 and Figure 4 is too low

Reviewer 2 Report

Comments and Suggestions for Authors

1. There must be something wrong with the Simple Summary and Abstract. you should check it and resubmit.

2. Although the author's research is very comprehensive, it is limited by the regional nature of the results and is difficult to conduct more in-depth research, which is not sufficient for publication as an ARTICLE. I suggest the author to modify the paper format to a short communication.

Reviewer 3 Report

Comments and Suggestions for Authors

Suggestions and questions:

Simple Summary:

L9-L10 “Such crossbreeding is used to create new breeds by selecting only individuals acquired from their parents' genes leading to higher performance in a particular performance trait.” It needs to be edited for a better understanding.

Abstract: Missing

Keywords: Adequate

Introduction:

L38 Add „García-Ruiz et al.“ To  „[7] stated,“

L 41    Check and edit: Thompson et al. and  Gorelik et al. [9,10] found a significant …

L42-L44 These sentences need to be edited for better understanding.

„The greatest production losses occurred early in life and in the early stages of lactation. Reduced productive life had probably a greater negative impact on dairy farm economics than production losses. The duration of productive life is higher in outbred and inbred cows with a moderate 45 level of inbreeding“

Materials and Methods

Could authors explain why cows “that reached more than 5 lactations” or “milk yield of less than 1700 kg“ are considered censored?  (L89-90).

Results

L139 The mean inbreeding coefficient was calculated separately for inbred cows (n = 161,919) in the base set. Could you explain why you did not use the non-inbred cows for the calculation of mean inbreeding coefficient?

Discussion

L210 An average inbreeding coefficient of 1.10% was calculated in the Slovak Simmental inbreed animals (Table 1), which is relatively low compared to other authors.

Could you explain the reasons for the low inbreeding coefficient in Slovak Simmental?

Conclusions

In conclusion, you stated, “The effect of inbreeding on cow survival was the smallest of all the factors evaluated, “ but this suggestion needs to be more explained. For example, the effect is statistically significant; why is it necessary to take into account inbreeding, and so on?

Round 2

Reviewer 2 Report

Comments and Suggestions for Authors

The author has done an excellent job of revising the paper and has persisted in publishing the results in article form. Please consider the suggestions from other reviewers when deciding whether to revise this into a short communication. I do not have any specific opinions or suggestions at this time.